# Evaluating the Global State of Ecosystems and Natural Resources: Within and Beyond the SDGs

**Christopher Dickens** [1,*] , **Matthew McCartney** [1] , **David Tickner** [2] , **Ian J. Harrison** [3] ,
**Pablo Pacheco** [4] and **Brown Ndhlovu** [1]

[1] Sustainable Water Infrastructure and Ecosystems, International Water Management Institute (IWMI), 127, Sunil Mawatha, Battaramulla, Colombo 10120, Sri Lanka; m.mccartney@cgiar.org (M.M.); brownkayirinde@gmail.com (B.N.)

[2] Chief Freshwater Advisor, World Wildlife Fund (WWF-UK), Living Planet Centre, Woking GU21 4LL, UK; DTickner@wwf.org.uk

[3] Moore Center for Science, Conservation International, Arlington, VA 22202, USA; iharrison@conservation.org

[4] Global Forest Lead Scientist at World Wildlife Fund (WWF), Washington, DC 20037, USA; pablo.pacheco@wwf.org

* Correspondence: c.dickens@cgiar.org; Tel.: +94-76-997-8752

**Abstract:** The Sustainable Development Goals (SDGs) purport to report holistically on progress towards sustainability and do so using more than 231 discrete indicators, with a primary objective to achieve a balance between the environment, social and economic aspects of development. The research question underpinning the analyses presented in this paper is: are the indicators in the SDGs sufficient and fit for purpose to assess the trajectory of natural resources towards sustainability? We extracted the SDG indicators that monitor the state of natural resources, or alternately support policy or governance for their protection, and determined whether these are adequate to provide the essential data on natural resources to achieve the aims of the SDGs. The indicators are clustered into four natural resource categories—land, water (both marine and freshwater), air and biodiversity. Indicators for monitoring land resources show that the most comprehensive land resource indicator for degraded land is not fully implemented and that missing from land monitoring is an evaluation of vegetation health outside of forests and mountains, the condition of soils, and most importantly the overall health of terrestrial ecosystems. Indicators for monitoring water resources have substantial gaps, unable to properly monitor water quality, water stress, many aspects of marine resources and, most significantly, the health of fresh and salt water ecosystems. Indicators for monitoring of air have recently become more comprehensive, but linkage to IPCC results would benefit both programs. Monitoring of biodiversity is perhaps the greatest weakness of the SDG Agenda, having no comprehensive assessment even though narrow aspects are monitored. Again, deliberate linkages to other global biodiversity programs (e.g., CBD and the Post-2020 Biodiversity Framework, IPBES, and Living Planet) are recommended on condition that data can be defined at a country level. While the SDG list of indicators in support of natural resource is moderately comprehensive, it lacks holistic monitoring in relation to evaluation of ecosystems and biodiversity to the extent that these missing but vital measures of sustainability threaten the entire SDG Agenda. In addition, an emerging issue is that even where there are appropriate indicators, the amount of country-level data remains inadequate to fully evaluate sustainability. This signals the delicate balance between the extent and complexity of the SDG Agenda and uptake at a country level.

**Keywords:** sustainability; Sustainable Development Goals; SDG; resource security; land; water; air; biodiversity

## 1. Introduction

There is a tension between the availability of natural resources and conventional pathways to socio-economic development. In order to develop, society needs to exploit natural resources (land, water, air, biodiversity, etc.) and may do so in ways that are entirely sustainable, i.e., where the resource is fully replenished usually by natural processes, or may do so where sustainability is compromised by short-term exploitation [1,2]. The current trends in population growth, changing lifestyles, consumption patterns (including overconsumption with diets relying on a narrow range of crops and livestock), advancing development, and economic activities are ramping up pressures on natural resources which are showing signs of stress, increasing the risk of collapse of natural ecosystems and associated loss of essential services [3–5]. Climate change adds further to this stress, with expanding developments using fossil fuels leading to greater carbon emissions, which in turn are altering weather and rainfall patterns and further exacerbating stress on natural resources.

Historically, there have been unique cases where natural resources were excessively depleted, resulting in the almost complete collapse of society, for example Easter Island [6] and the Mayan civilization [7]. More recent history is showing that we as a society are using resources well beyond the capacity of the globe to provide. Thus, the Global Footprint Network [8] calculates that we currently use 1.7 planets to support humanity's demand on Earth's ecosystems, suggesting that society is using up its reserves and likely to be compromising its future. The Planetary Boundary concept [9] estimates that four out of the nine natural resource thresholds, that should not be crossed, have already been transgressed [10], while other assessments suggest that this is an underestimate as water resources should also be recorded as transgressing, as already 80% of the world's population live in areas where there are high levels of threat to water security [11] and water crises are now ranked as third in the top 10 global risks to the world economy [12]. At a planetary level, the WWF's Living Planet Report demonstrates major declines across many indicators that represent natural resources [13], while a recent paper by Albert et al. [14] documents the crisis in detail. That the evidence is becoming plain even at a global level is illustrated by the scourge of climate change [15], degraded lands [13], degraded water [16] and collapsing biodiversity [17], all of which are associated with degradation of natural resources.

Sustainable development is reliant on the provisions of nature to meet social and economic developmental needs [5,18,19]. The final output of the Rio + 20 Conference called for "*protecting and managing the natural resource base for economic and social development*" [20]. This relationship was well described in the concepts of ecosystem services by the MEA [21] and is now taken further by the IPBES [17] and the Convention on Biological Diversity draft Post-2020 Framework [22] with the concept of "Nature's contributions to people", which directly establishes the context of the relationship between natural resources and sustainable development. Monitoring and management of what remains of natural resources have thus become vital for ensuring a sustainable future.

Agenda 2030 on the Sustainable Development Goals [23] has become a prominent global response to the pressing issue of declining sustainability. The 17 goals, 169 targets and now >240 indicators are designed to indicate the progress of the world towards sustainability, and deliberately cover the three pillars of sustainability, i.e., environment, social and economic [24] that were promoted during the Rio Summit (Agenda 21 Chapter 8.4). The overriding concept is that sustainability can only be attained by balancing these three dimensions in an integrated and holistic way [25]. This balance introduces the idea that where synergistic outcomes for people and nature are not available, trade-offs need be made between the three in order to reach a sustainable balance. There is increasingly a rejection of the concept of a dichotomy of environment vs. development, moving instead to perspectives such as ecosystem based engineering and green infrastructure, which bring ecosystems into development options as part of the solution, emphasizing that there should be no opposition [26].

Emerging from Rio + 20 and before the SDGs were published, UNEP set up an International Resource Panel to promote incorporation of natural resources into the SDGs [27] which went as far as to recommend inclusion of a resource dedicated SDG (which did not happen) and noted that

"sustainable resource management" is imperative for human well-being and for sustained economic development. UNEP [3] went on to say that "*natural resources management. is a prerequisite to achieving sustainability, otherwise the SDGs will not fulfil their fundamental purpose of ending extreme poverty by 2030 and addressing all aspects of sustainable development*". The UN Agenda 2030 on the SDGs [23] states "*We recognize that social and economic development depends on the sustainable management of our planet's natural resources*". It thus recognizes the interrelationships between human well-being, nature and economic growth. However, Hutton et al. [28] believe that the SDGs are excessively loose in their formulation and are vague in how the ideals they represent should be realized and translated into pathways for development. This includes the quantification of the indicators, as the SDG targets and indicators are deliberately non-prescriptive about their achievement.

Although UNEP [3] praised the SDG Agenda [23] for its acknowledgement of the links between environment, social and economic issues and the need for a balance between all three to achieve sustainability, there are indications that all is not well in this regard and that SDG progress reports are dominated by social and economic development issues with little attention given to protection of natural resources [18,29].

Reporting on progress, UN-DESA [30] noted that while all of the five SDG 15 (land) indicators of society's response show positive trends, the indicators that show the state of life on land both indicate declines. They question why the overall state of nature, and by implication, natural resources, is declining despite increasing efforts towards conservation and sustainable development, which should be an urgent priority if SDG 15 is to be met. The UN Environment Assembly [31] made a submission to the High-Level Political Forum of the UN (HLPF) noting that most of the indicators for which good progress is being made relate to policy, improved reporting or increased funding efforts as opposed to a positive change in the state of the environment itself. They note increases in terrestrial, mountain and marine protected areas; efforts to combat invasive species; use of renewable energy; sustainability reporting and mainstreaming in policy; and development assistance for climate change and the environment. However, then they note that there is too little data to assess the bulk of the environmental indicators including ocean acidification (targets 14.1 and 14.3), water quality (target 6.3), water stress (target 6.4) and mountains (target 15.4). It could be envisaged that improvements in the state of the environment and natural resources will lag enhanced efforts and changes in policy, with the latter being a pre-requisite for the former. Nevertheless, lack of progress to date is a concern, as is the question of whether or not sufficient information is being obtained. UNEP [32] suggest that the SDG indicators will not provide all the information needed to understand the health status of the planet. Another report by UN Environment to the HLPF in 2020 shows that only 20% of countries reported biodiversity as a priority in their voluntary national reviews, again suggesting a dearth of information or lack of interest [33].

This paper addresses whether there are adequate goals, targets and indicators to provide the evidence needed for protection and management of natural resources. The authors have used the definition of natural resources as "*raw materials occurring in nature that can be used for economic production or consumption: they are subdivided into four categories: mineral and energy resources, soil resources, water resources and biological resources*" [34]. The SDG Agenda Item 33 names natural resources as "*oceans and seas, freshwater resources, as well as forests, mountains and drylands and to protect biodiversity, ecosystems and wildlife*" [23]. Of note is that neither of the above two definitions list air as a natural resource, a serious omission given the advent of climate change. Four categories of natural resources have thus been considered here—land, water, air and biodiversity.

## 2. Materials and Methods

In order to assess the inclusion of natural resource protection within the SDGs, all 231 unique indicators of the Agenda 2030 were assessed. The list was that developed by the Inter-Agency and Expert Group on SDG Indicators (IAEG-SDGs) and published in the "Global indicator framework", adopted by the General Assembly (A/RES/71/313) with a summary list (E/CN.3/2020/2) published in

March 2020. The key source of data was the metadata publication of the UN Statistical Commission [35] that is continuously updated as indicators are developed. Data were gathered by seeking out those indicators that provide one of two perspectives related to resource protection; firstly a measure of some aspect of the natural resources, that ensures that the resource is evaluated quantitatively thus enabling awareness of how much remains (e.g., how many hectares, or $km^3$ or the size of the fish population etc.); and secondly those indicators that evaluate measures that directly support protection of the quantity of resources (e.g., policy that supports protection of soil resources, or management to curb water withdrawals etc.). Each indicator and its contribution towards monitoring natural resources was thus evaluated and ranked, where (i) the indicator directly and quantifiably (in $km^2$ or $Mm^3/a$ or mg/L, etc.) monitors a natural resource; (ii) the indicator measures the conditions for natural resources protection (e.g., natural resource protection policy); and (iii) the indicator does not include either of the above and thus is purely social, economic or development oriented.

## 3. Results

Figure 1 shows that 18 or 7.8% of the 231 unique indicators monitor natural resources directly (see Table 1), 19 or 8.2% monitor conditions for natural resource protection but do not directly quantify them (see Table 2), while 194 or 84.0% of indicators measure socio-economic development and governance. This does not, however, mean that natural resources are not adequately covered.

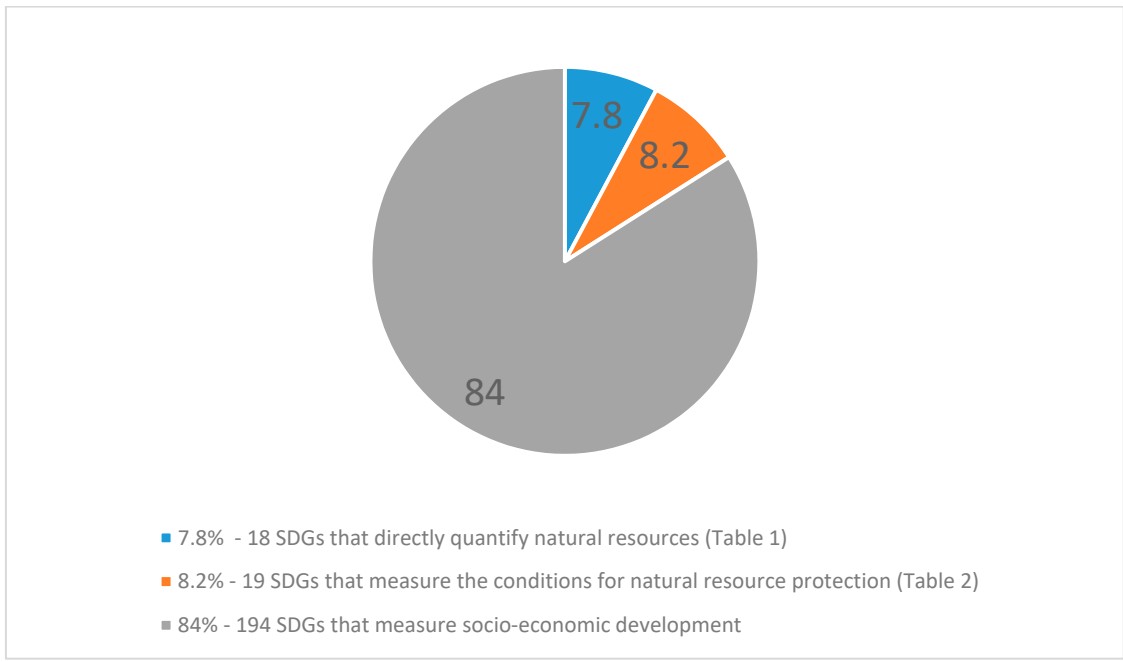

**Figure 1.** SDGs indicators' monitoring of natural resources against monitoring of social and economic development, based on 231 unique indicators.

**Table 1.** SDG indicators that directly measure a natural resource, and likely gaps in natural resource monitoring. This list is based on the global indicator framework adopted by the General Assembly (A/RES/71/313) E/CN.3/2020/2 published in March 2020.

| | Indicators | Natural Resources Directly Monitored/Measured | SDG Gaps in Data and Information |
|---|---|---|---|
| | | Land | |
| 2.4.1 | Sustainable agriculture | Eleven sub-indicators but only soil health is a direct measure of a natural resource | Water resources for agriculture |
| 11.3.1 | Land consumption rate | Areal extent of newly developed urban land (indicating loss of land) | Consumption of non-urban land |
| 15.1.1 | Forest area | Forest as a percentage of total land area | Forest condition (deforestation and forest restoration) by type of forest |
| 15.3.1 | Degraded land | Land cover, net primary production of vegetation and carbon stock | Stages of land degradation/erosion Soil condition, nutrients and fertility Salinization and desertification |
| 15.4.2 | Vegetation cover of mountains | Green (chlorophyll) land cover at altitude classified as mountains | Vegetation cover of flat lands and wetlands |
| | Other land resource gaps | | Other natural ecosystem conditions (e.g., grasslands, savannah, wetlands) Ecosystem health of land |
| | | Water (freshwater and marine) | |
| 6.3.2 | Ambient water quality | Limited water quality parameters expanding with progressive monitoring | Multiple water quality parameters depending on progressive monitoring Biomonitoring data |
| 6.4.2 | Water stress | Total fresh water quantities (withdrawn, renewable surface and groundwater resources and environmental requirements or e-flows) | Does not illustrate change over seasons |
| 6.6.1 | Spatial extent, quantity and quality | Spatial extent (lakes, rivers, estuaries, artificial water bodies, vegetated wetlands), water quality (chlorophyll and total suspended solids) and volume of water discharge in rivers and estuaries and a measure of groundwater depth | Quantities of ice/snow, soil water, water in vegetated wetlands Aquatic ecosystem types Natural vs. artificial wetlands Natural vs. artificial water bodies Groundwater volumes Health of ecosystems |
| 14.1.1 | Coastal eutrophication and plastic | Eutrophication (chemical; algae and biodiversity but presently only chlorophyll as proxy) and plastic debris although only beach litter presently | Other nutrient pollutants Turbidity Marine ecosystem health |
| 14.3.1 | Marine acidity (pH) | pH, DIC (dissolved inorganic carbon), $p\text{CO}_2$ (carbon dioxide partial pressure), and TA (total alkalinity) | |

**Table 1.** *Cont.*

| | Indicators | Natural Resources Directly Monitored/Measured | SDG Gaps in Data and Information |
|---|---|---|---|
| | | Other water resource gaps | River connectivity<br>Ocean water quality<br>Aggregate extraction<br>Linkage between SDG6, 14 and 15 |
| | | Air | |
| 11.6.2 | Particulate matter | Fine suspended particles in the air (in urban areas) | Nitrogen dioxide, sulfur dioxide, other pollutants |
| 13.2.2 | Greenhouse gas | Total greenhouse gasses emitted per year | (method under development) |
| | | Other air resource gaps | Air temperature<br>Climate change over time |
| | | Biodiversity | |
| 2.5.1 | Genetic resources | Plant and animal genetic resources of potential or actual value for agriculture | Plant and animal genetic material of non-agricultural species |
| 14.4.1 | Fish stocks | Fish catch, yield and production of commercial species, abundance compared to sustainable yield | Freshwater fisheries<br>By-catch<br>Marine and freshwater biodiversity |
| 15.1.2 | Measure of protected areas | Terrestrial and freshwater protected areas that are protecting a limited number of important species dominated by birds, plus endangered species | State/health of these protected areas<br>Limited inclusion of freshwater biodiversity<br>Environmental flows carried from 6.4.2 |
| 15.2.1 | Sustainable forest management | Forest area and biomass with aspects of biodiversity | Sustainable non-forest management<br>Forest biodiversity |
| 15.4.1 | Mountain biodiversity | Key biodiversity mountainous areas (no species data), dominated by birds, plus endangered species | State/health of mountains<br>Flatland, floodplain, wetland and delta biodiversity<br>Mountain biodiversity |
| 15.5.1 | Red List Index | Limited number of species in each Red List Category | State of the majority of global species not listed (millions) |
| | | Other biodiversity gaps | Biodiversity data across all ecosystems<br>Biodiversity indices |

**Table 2.** SDGs Indicators that monitor conditions for natural resource protection but do not directly quantify them.

| | Indicators that Support Natural Resource Protection | Natural Resource that May be Supported |
|---|---|---|
| | Land | |
| 15.2.1 | Progress towards sustainable forest management | Forest and land area |
| | Water freshwater and marine | |
| 6.3.1 | Proportion of wastewater safely treated | Water quality |
| 6.5.1 | IWRM implementation | Water quantity, quality, ecosystems |
| 6.5.2 | Transboundary basin area with arrangement for water cooperation | Water quantity, quality, ecosystems |
| 14.2.1 | Countries manage marine and coastal areas using ecosystem-based approaches | Water quality, biodiversity, marine ecosystems |
| | Air | |
| 7.b.1 | Investments in energy efficiency | Air quality, natural resources |
| 13.2.1 | Climate change policy to adapt and lower greenhouse gas emissions | Air quality and climate change |
| | Biodiversity | |
| 11.4.1 | Expenditure on natural heritage | Biodiversity, ecosystems, features |
| 14.5.1 | Coverage of marine protected areas including key biodiversity areas | Marine ecosystems, biodiversity |
| 14.6.1 | Instruments that combat illegal fishing | Fish stocks and thus commercial species |
| 14.7.1 | Sustainable fisheries | Fish stocks |
| 14.c.1 | Sustainable use of the oceans and their resources | Fish stocks and thus commercial species, biodiversity, water quality, marine ecosystems |
| 15.8.1 | Prevention or control of invasive alien species | Biodiversity and ecosystems |
| 15.9.1 | National biodiversity values in accordance with Aichi Biodiversity Target 2 (biodiversity inventory and valuation) | Biodiversity and economic value |
| 15.a | Finance to conserve and sustainably use biodiversity and ecosystems | Biodiversity and ecosystems |
| 15.b | Finance to conserve and restore forests | Forest biodiversity and extent |
| | All resources | |
| 12.4.1 | Agreements to reduce hazardous waste | Air, water, soil |
| 12.4.2 | Proportion of hazardous waste treated | Air, water, soil |
| 12.b.1 | Tourism impacts on the environment | Environment |

The SDG indicator methods are in a continual state of development, categorized by the IAEG into Tier 3 (under development), Tier 2 (ready but not operational) and Tier 1 (fully operational in many countries). According to the IAEG in 2020 [36], the tier status of all of the indicators in Tables 1 and 2 were either Tier 1 or 2 and thus all were available for implementation.

Tables 1 and 2 have been structured following guidelines of ISO, which divides indicators into three categories—environmental condition indicators (ECIs) which are in Table 1, and operational performance indicators (OPIs) and management performance indicators (MPI), which are in Table 2.

The information in Table 1 reflects only those indicators that directly and deliberately set out to quantify some aspect of the natural resource, and are thus a measure of environmental condition (ECI) and may be affected by over-utilization. In some indicators, this may be included in an index where the natural resource measure may be obscured (e.g., 15.4.2—the Mountain Green Cover Index), the indicator thus requiring disaggregation.

The indicators presented in Table 2 do not themselves contain any quantification of natural resources and thus are not able to reflect their actual status. However, these indicators may provide a basis for protection of natural resources by providing enabling conditions. For example, indicator 14.6.1 does not monitor any fish species but rather country implementation of the Code of Conduct for Responsible Fisheries and related instruments (CCRF) which encourages the conservation and protection of fish stocks through combating illegal fishing and over exploitation. In the same manner, indicator 6.5.1 does not monitor any water quality or quantity parameters but by monitoring country commitments to implementation of policies, laws, institutional arrangements, budgeting and financing and strategies for water resources development, protection of water quality may be achieved. Table 2 is thus a mix of operational performance indicators (OPIs) and the management performance indicator (MPI).

## 4. Discussion

According to UN Environment, 86 out of the 169 SDG targets directly or indirectly seek to reduce environmental damage or emphasize the critical role of natural resources and ecosystem services in

ensuring human well-being and prosperity, [37]. However, these cover a wider scope than just natural resources. According to UNEP, the Stockholm Resilience Centre categorized the SDGs into those that support the biosphere (6,13,14,15), society (1,2,3,4,5,7,11,16) and economy (8,9,10,12), [38].

This paper goes beyond goals and targets to the indictor methods themselves, from which it is possible to record a total of 30 targets and 37 indicators (Tables 1 and 2), comprising methods directly quantifying natural resources (18 indicators, 7.8% of total—Figure 1) or providing supporting conditions for natural resource protection (19 indicators, 8,2% of total). These constitute a small percentage compared to the balance of the indicators that are focused on socio-economic and governance issues (84% of total—Figure 1). These results are similar to those produced by Wackernagel et al. [19], who relied on a draft list of indicators and argued that they were dominated by indicators related to social development, while those related to resource security receive substantially less attention. This does not mean, however, that these are not sufficient as, in theory, a few comprehensive natural resource indicators could provide adequate information for management to ensure sustainability. By contrast, the Convention on Biological Diversity (CBD 2020) [39] has drafted a list of some 161 indicators for monitoring of biodiversity issues alone, which themselves have been challenged as inadequate, suggesting that the SDG attempts to be brief may be limiting.

Table 1 presents possible gaps or omissions from the SDG indicators. Compiling such a list is fraught, as it is clear that the SDG process was designed to be as simple as was necessary to represent sustainable development. However, this paper has the objective of highlighting the shortcomings of SDG indicators to inform any potential upgrade to the SDGs over the next years, and in addition, to motivate research efforts that may be directed to infilling the gaps. The paper also sets out to indicate the types of data that are missing in order to highlight the limitations of the SDG indicators and the care that is needed in interpretation. It would be possible to conduct a comprehensive gap analysis by comparison of the SDG indicators with the best that the world has to offer, where each SDG indicator is comprehensively monitored for omissions, but such an exercise is of limited value as implementation of such a comprehensive program would not be purposeful for the SDGs and would be beyond the capacity of most countries to implement. Take water quality for example. There are numerous recommended guidelines for environmental water quality monitoring that are not directly used in indicator 6.3.2 and which are detailed under the heading Water Resources below. However, the SDGs are by nature pragmatic, their strength coming from their wide-span including social, economic and environment indicators, together with a regular update to establish trends. Key also is their almost universal uptake, high-level publicity and support. The summary of the natural resources monitored by relevant SDG indicators (Tables 1 and 2) show varied levels of support for monitoring of natural resources; a summary and analysis is provided below. Note that all references to SDG indicator methods can be made to UNSTATS [35] unless otherwise stated.

### 4.1. Land Resources

Tables 1 and 2 show the extent to which land resources are monitored within the SDGs. This includes the health of agricultural soil (as part of sustainable agriculture, indicator 2.4.1), urban land conversion (11.3.1), forest area (15.1.1), greening of mountains (15.4.2), and the area of degraded land (15.3.1). Regulation of hazardous waste (12.4), and management of tourism would benefit land resources (12.b.1). Gaps in the monitoring include the consumption of rural land (11.3.1) the health or state of forests (15.1.1), vegetation cover of flat lands and wetlands (15.4.2) as well as many nuances about degraded land including the stages of land degradation, soil condition, nutrients and fertility, salinization and desertification (15.3.1). A major gap in all monitoring here is a measure of ecosystem health of landscapes.

Some of the indicators that are included for land are affected by the lack of precision of concepts associated with them (e.g., forest, degradation, productivity) which adds uncertainty to the outputs. Most of the indicators are multidimensional which results in complex indices that may contain, embedded within them, quantifiable measures of land resources (e.g., land degradation,

forest condition). The SDGs rely on already agreed conceptual frameworks to measure some of these indicators when they are available. Thus, for example, while there is no agreed framework for forest condition, there is one to measure land degradation neutrality (LDN) [40] that forms part of 15.3.1.

Indicator 15.2.1 [35] centers on implementation of sustainable management to halt deforestation, restore degraded forests and substantially increase afforestation and reforestation. While there are agreed definitions of sustainable management, deforestation and afforestation/reforestation, restoration is a more elusive concept since there are many different practices leading to forest restoration (e.g., passive natural restoration, active restoration, tree plantations), which may also overlap with afforestation and reforestation depending on where forest restoration takes place, and the time frames considered [41]. Forest degradation is more difficult to measure since results from changes within forests that negatively affect the structure or function of the stand or site over many decades, can thereby lower the capacity to supply products and/or ecosystem services [42]. There are different types of forest degradation including, among others, the ecological quality and integrity of forests [43].

Indicator 15.3.1 is one of the key measures of land condition in the SDG framework documenting land that is in any way degraded, although the definition of "land degradation" confusingly applies only to "rain fed cropland, irrigated cropland, or range, pasture, forest and woodlands", by implication only land used for productive purposes and assuming that other land is not subject to degradation. This definition and method are aligned with that of the United Nations Convention to Combat Desertification (UNCCD) [40]. The conceptual framework for this indicator is the Land Degradation Neutrality status (LDN) [40], which contains three sub-indicators—land cover (land cover change), land productivity (net primary productivity) and carbon stocks (soil organic carbon)—which together are regarded as reasonable proxies for changes in land based natural capital which are universally acceptable and applicable [40], though it must be recognized that some land in a "natural" condition may not always be productive or have a high carbon stock. The sub-indicators do, however, monitor all forms of land and the method focusses on the trends of change to indicate degradation. One of the sub-indicators, "land cover", is essentially a measure of vegetation, water bodies and human development, and is used as a proxy to reflect the capacity to deliver land based ecosystem services, and thus suggest land in good condition. The Good Practice Guidance for Indicator 15.3.1, [44] states that the component sub-indicators are "*necessary but not sufficient*", acknowledging that they do not cover all possible perturbations. The indicator method [33] documents 19 different forms of land degradation that could be but are not all included in monitoring. They acknowledge that not all available indicators could be included in the index as this would become too complex (including landscape modification, soil erosion and compaction, salinization, acidification, fertility, contamination, soil extraction, aridification, vegetation cover, vegetation community functioning, biomass, biodiversity, seedbank, weeds, invasive species, habitat, hydrological modifications, and groundwater).

Gaps in some of the other indicators are also apparent. Thus, for indicator 15.1.1, the forest area index, the index itself recognizes that it is "*a rough proxy for the extent to which the forests in a country are being conserved or restored, but it is only partly a measure for the extent to which they are sustainably managed*". With regards to target 15.2, there is a lack of indicators linked to forest degradation and forest condition, as well as others able to capture forest cover change dynamics (e.g., fragmentation, afforestation, reforestation and restoration). SDG 15.4.2 on the green cover of mountains states that there is a "*direct correlation between the green coverage of mountain areas and their state of health, and as a consequence their capacity of fulfilling their ecosystem roles*". This statement should not go unchallenged in that that some healthy natural areas do not have green cover (e.g., deserts, very high mountain ranges). An important omission from all of the indicators on the land resource, and despite the best intentions of 15.3.1, is that there is no general measure of ecosystem condition.

## 4.2. Water Resources

Table 1 shows that water quality is covered to a limited extent (6.3.2 and 6.6.1); water quantity, including inventories of renewable resources and environmental flows (6.4.2) are supported by the

direct measure of water volumes and discharge (6.6.1). The latter indicator also documents the change in spatial extent of water bodies and wetlands, and in its latter phases monitors groundwater as well. Freshwater resources also appear in 15.1.2 that sets out to monitor the proportion of important ecosystems, including freshwater, that are protected, but this indicator does not measure the water resource directly. Coastal and marine water resources include indicators of eutrophication, plastic pollution (14.1.1) and acidification (14.3.1).

Possible gaps in water quality monitoring (6.3.2) are the wider range of water quality variables that are possible but not specified; the water stress (6.4.2) indicator does not focus on the times of year when stress is important, as the annual average water stress is not often relevant to stakeholders; there is undocumented water quantity in snow and ice, soil water and in vegetated wetlands (6.6.1). The health of freshwater ecosystems is not monitored, a major omission, while for marine ecosystems this is done only by proxy. An important component of river ecosystems that is missing is the connectivity of rivers, especially when disrupted by dam placement (6.6.1). This could now be addressed by applying the newly developed river connectivity index (Grill et al. 2019). In coastal and marine systems the SDGs take a light touch, giving emphasis to nutrients and eutrophication, but providing little detail on ecosystem state (14.1.1.).

There are a number of other indicators, which if implemented could contribute to protection of water resources (Table 2). Increased treatment of wastewater would protect water quality (6.3.1); IWRM implementation and transboundary governance of water resources should, if properly implemented with sustainable goals in mind, protect many aspects of the resource (6.5); application of ecosystem-based approaches to management of marine and coastal areas should protect all aspects of the resource (14.2.1); reduction in hazardous waste would also protect water resources (12.4).

Monitoring water quality trends (6.3.2) is widely recognized for its complexity [45]. Because it is context specific, it remains a challenge; there are a large number of variables that influence water quality [46]. SDG 6.3.2 in its foundation level of progressive monitoring is limited to only five variables (oxygen, nitrogen, phosphorus, conductivity, pH). This short list is pragmatic, as it was assessed that most countries would not be in a position to provide more comprehensive monitoring. The short list is also designed to be manageable with only basic field equipment making this globally more appropriate [47]. This list does, however, divert substantially from other recommendations for general water quality monitoring which are much more extensive such as Bartram et al. [48], CCME [49], Standard Methods [50], UNICEF [51], USDA [52], (ISO) [53], WHO [54] and the GEMStat program [55], with the SDG indicators avoiding the associated complexity. A similar challenge was faced by the World Water Quality Assessment [56] which is also not comprehensive in its inclusion of variables. There are attendant risks in not monitoring comprehensively, for example emerging pollutants of water have been identified as one of the greatest threats to biodiversity [57]. Countries implementing SDG 6.3.2 have the facility to extend the monitoring to locally relevant variables, but the challenge for the UN will be how to deal with dissimilar data sets emerging from each country which will challenge global reporting. 6.6.1 also contributes to water quality by adding earth observation data on chlorophyll and suspended solids or turbidity of lakes. However, this is confined to large ecosystems and artificial reservoirs, so does not provide a comprehensive picture of the state of aquatic ecosystems.

Indicator 6.4.2, the "water stress indicator", has been previously described [58] and provides measurements of what water remains in the environment for use, and includes consideration of environmental flows i.e., the quantity and timing of flows required to protect ecosystems so that they can continue to provide benefits to society. The environmental flow data are, however, hidden within the index and needs to be disaggregated if the data are to be available as a descriptor of river ecosystem health. Ideally, these data should be directly linked to 6.6.1 on water-related ecosystems where they can usefully be used to report on the condition of aquatic resources. Environmental flows can also usefully be used as an indicator for aquatic protected areas for 15.1.2. Reported as an annual figure, the 6.4.2 indicator also does not show the variation in water stress as affected by seasonal changes, which would be of greater interest at a local level, especially in river basins where there is substantial

natural variability in hydrology and/or pronounced seasonal variability in water use. Indeed, it has been shown that some 4 billion people live under conditions of water stress for at least one month of the year [59].

In terms of spatial extent of water bodies, the 6.6.1 indicator method is globally comprehensive as it is based on earth observation methods which are developing in tandem with the implementation of this method [60]. Separation of artificial (e.g., dam reservoirs) from natural open water bodies is intended in the method but must be carried out otherwise estimations of extent are meaningless. Inclusion of small wetland ecosystems remains a challenge if they are less than a quarter hectare in extent, which in some ecosystems is the predominant size, e.g., on steep mountainsides. Inclusion of temporary systems that do not demonstrate either open water or dramatic vegetation changes is also a challenge. Assessment of vegetated wetlands, however, remains at the edge of capability of the developing methods so these data remain elusive and there is also little information on the spatial extent of ecosystem types, which may obscure changes in spatial extent that happen from one type to another with a subsequent loss of biodiversity or ecosystem services [61]. Separation of natural from artificial wetlands (e.g., rice paddies) is also vital for proper understanding. Water quantity changes are well covered in terms of open water (discharge and spatial extent) and to a lesser extent groundwater, but there is concern over the poor state of global data as hydrometric networks are in decline [26]. New earth observation methods could, however, soon take over hydrological monitoring [62]. There is also little information on the temporal (seasonal and interannual) fluctuations of flow which are so important for sustainability of aquatic ecosystems [63].

The coastal and marine resource indicators have a primary focus on monitoring national waters [38], and are supported by multiple international practices and conventions, which have only been partly synthesised to provide relevant indicators for the SDGs. In addition to the indicator methods provided [35], UN Environment have produced comprehensive supporting documents for 14.1.1, 14.2.1 and 14.5.1 [38], with further support pending. These reports describe that the anticipated indices are not yet ready for implementation, but in the meantime chlorophyll-a concentration is the proxy indicator for eutrophication (14.1.1); "beach litter" is the proxy indicator for marine plastic litter (also 14.1.1); and Regional Seas Coordinated Indicator 22 "Integrated Coastal Zone Management (ICZM) protocols" as proxy indicator for ecosystem-based management in coastal zones (14.2.1). These proxy indicators are in line with the Regional Seas Conventions and Action Plans agreed at their 18th Global Meeting in 2016 [38].

The intention is that SDG 14 indicators (and proposed proxies) measure the state and quality of the impacted ecosystems, rather than the drivers and pressures underlying these drivers. What will be delivered in future will include, for 14.1.1 nitrogen and phosphorous concentrations, as well as silica as this would allow interpretation of harmful algae growth, chlorophyll, biomass and turbidity. Dissolved oxygen is also to be used as an indicator. The Regional Seas Programs that dominate developments in this line have three categories of eutrophication indicators, nutrients, direct effects such as phytoplankton, and indirect effects such as oxygen, organic carbon, zooplankton and fish. However, the 14.1.1 index will only be fully developed by 2021.

The SDG and proxy indicators only capture part of the associated SDG targets. In the long-term, these limitations will have to be addressed to ensure that SDG 14 is fully met [38]. Deliberate linkages between fresh and marine SDGs would also provide greater understanding as these ecosystems are closely entwined, with rivers being a major source of pollutant and sediment input to the sea and many species (especially fish species) migrating between the two systems.

A major limitation of the freshwater reporting is the lack of any comprehensive measure of ecosystem health, while the marine indicators presently rely on chlorophyll and biomass but do not show more detail. Closely related to this, is the lack of information on the extent of different ecosystem types or habitats in both freshwater and marine areas; information which could be used as a foundation of the biodiversity evaluation to come. In marine areas this would include monitoring of individual marine and coastal habitats, such as coral reefs, seagrass, saltmarsh and mangroves (although the extent

of these is captured in 6.6.1). Closely related to ecosystem health is the monitoring of environmental flows (e-flows), which while forming part of the 6.4.2 indicator [64], have no goal or objective for their implementation as a measure of progress towards ecosystem health. In addition, there is a lack of information on the discontinuity of ecosystems [65] that have been divided by development, e.g., by construction of dams. This discontinuity can provide a break in the entire functioning of river ecosystems, by disruption of sediment transfer, by changing water quality, by obstructing movement of fauna in both directions, by altering flooding regimes and subsequent scouring of substrates and the riparian zone etc.

### 4.3. Air Resources

Initial monitoring by the SDGs was limited to particulate matter PM2.5 (11.6.2—level of fine particulate matter in the air of cities), but in 2020 indicator 13.3.2 was introduced to monitor total greenhouse gas emissions (Table 1), thus making a link to climate change although the method is not yet available [35]. PM2.5 is a key global pollutant that results from fires and engine emissions and is a key health issue. A global study of 652 cities on the impacts of PM2.5 and PM10 provided evidence of a positive association between short-term exposure to PM10 and PM2.5 and daily all-cause, cardiovascular, and respiratory mortality [66]. A more comprehensive monitoring program, the WHO air quality monitoring guidelines [67], recommended monitoring of particulate matter (PM), ozone ($O_3$), nitrogen dioxide ($NO_2$) and sulfur dioxide ($SO_2$). The combined 11.6.2 and 13.2.2 will cover much of this recommendation.

UN Environment with UN-HABITAT and IQAir in 2020 created a new database on air quality concentrating on PM2.5, the Urban Air Action Platform, [68] which will provide valuable support to the SDGs especially for countries lacking infrastructure. This has as its support the Global Environment Monitoring System for Air (GEMS Air), which is the UN Environment mechanism on air quality monitoring, which promotes monitoring of particulate matter of various sizes but also NO, $NO_2$ and $SO_2$.

The addition of SDG 13.2.2 on greenhouse gasses means that the SDGs now will align with the IPCC which will enable cross-linkages between climate change and sustainable development to be made. Still missing from the SDGs is a direct link to air temperature.

Air quality is at the center of the biggest environmental changes facing society due to climate change where air pollutants and temperatures are increasing year by year [15]. Air pollution is a publicly visible degradation of a natural resource with multiple examples that have made global headlines, e.g., the Australian bush fires of 2019–2020 [69], where fires and plumes of smoke coated the continent; air pollution in 2019 Delhi India was reported as a "Climate Emergency" [70]; the noticeably reduced air pollution in cities during the Covid−19 crisis, where concentrations of PM2.5, $NO_2$ and $SO_2$ in China decreased by 33.2%, 27.2% and 7.6%, respectively, compared with 2019 because of the slow-down in industry [71]. The WHO noted that policies to reduce air pollution offer a "win–win" strategy for both climate and health, lowering the burden of disease attributable to air pollution, as well as contributing to mitigation of climate change [72].

### 4.4. Biodiversity Resources

Table 1 shows a number of largely non-specific biodiversity indicators where minimal measures of biodiversity are contained within indicators with a wider objective. Thus, there is a measure of genetic resources for agriculture (2.5.1)—marine fish stocks, but without a measure of marine and freshwater biodiversity (14.4.1); the indicator of marine eutrophication has a requirement for evaluation of marine biodiversity but no detail or method direction is yet available (14.1.1); limited biodiversity in terrestrial protected areas (15.1.2); and limited biodiversity for forest and mountain area (15.2.1 and 15.4.1). The only dedicated biodiversity indicator is 15.5.1 but it is limited in scope. In all land, freshwater and marine environments, a full measure of biodiversity is missing.

There are a number of indicators which if implemented would provide protection to biodiversity resources but do not directly measure biodiversity (Table 2). These include indicators of policy and

financial support for protection of marine and land resources (14.4.1; 15a; 15b), marine protected areas (14.5.1), sustainable use of marine resources (14.6.1; 14.7.1; 14.c.1), trading of biodiversity (15.7.1) and limiting of alien species (15.8.1). Perhaps the most noteworthy is 15.9.1 that seeks to align the SDGs with Aichi target 2 (now addressed principally by post-2020 draft target 13), which provides inventories and associated values of biodiversity but the impact of the AICHI targets on the SDGs is not evident and was not mentioned in the UN SDG report of 2019 [73].

Biodiversity can be reported in a number of complementary approaches, using measures of biodiversity or of ecosystems with their component biodiversity, the spatial extent of important ecosystems or biodiversity areas, and the policies and management activities designed to give them protection. The SDGs make use mostly of the latter two approaches.

Goal 6 documents the extent of water-related ecosystems, but there is no measure of overall ecosystem condition or associated biodiversity. While the intention of the 6.6.1 target is to protect and restore water-related ecosystems, including mountains, forests, wetlands, rivers, aquifers and lakes, suggesting that some measure of biodiversity would be necessary, this has not been included in the final indicator. Early developments of the indicator included a measure of aquatic ecosystem health [74] but this was removed for the second round of data collection as it was concluded that most countries were not in a position to provide useful data. The option, however, remains recorded in the 6.6.1 method for future inclusion, with the ideal being a global dataset on aquatic ecosystem health that would include biodiversity. Freshwater fisheries are globally important and under threat [75,76], yet are a notable omission from the SDGs.

SDG 15 is established for protection, sustainable use and restoration of terrestrial ecosystems, and also to prevent biodiversity loss. SDG 15.1 is dominated by terrestrial ecosystems (especially forests and protected areas) but does include water ecosystems in its targets, including somewhat hidden in 15.1.2 the freshwater Key Biodiversity Areas [77,78], which are also found in 14.5.1 for marine ecosystems [79].

SDG 15.1.2 is a measure of the proportion of important sites for biodiversity covered by protected areas, including freshwater ecosystems. The indicator includes a measure of the biodiversity associated with these ecosystems, but the measure of biodiversity is limited in extent, and is dominated by birds and endangered species. It has been acknowledged that reporting on the number and extent of protected areas provides only a unidimensional measure of political commitment to biodiversity conservation and does not report on the effectiveness of biodiversity conservation [80], a perspective that has been recommended for resolution in the CBD Post-2020 evaluation. Further, as a tool to protect freshwater ecosystems, protected areas can only make a limited contribution because by nature most river ecosystems are mobile and cannot be contained within the boundaries of a protected area [81].

SDG 15.2.1 (sustainable forest management) considers the extent to which forests are incorporated into protected areas, and this includes forest management plans that in turn consider biodiversity aspects. Biodiversity data on forests, however, is not directly reported in the SDGs. Similarly 15.4.1 considers protected areas and mountains, the aim being to protect a fairly short list of important species. Species data are not directly obtainable from SDG reports.

SDG 15.5.1 is perhaps the most focused biodiversity indicator, using the IUCN Red List of Threatened Species that accounts for the risk to some 120,372 species [82] but does not document the status of the bulk of species (widely quoted as 8.7 million [83]), nor their interactions or changes in community structure etc. It was noted in IPBES [17] that indices such as the IUCN Red List categories are relatively coarse and may miss gradual declines of abundant, widespread species, which indicators based on species' abundances could capture [84]. Thus, while Goal 15 even includes in its title to "halt biodiversity loss", it provides only a limited direct measure of biodiversity [35].

The most widely quoted snap-shot evaluation of the state of the world's biodiversity is possibly the WWF/ZSL Living Planet Index (2018) that considers 16,704 populations with 4005 species and shows changes in the average populations of species. This same report and also Mace et al. [85] also promotes having more than one biodiversity indicator, and they present an additional three including

the IUCN Red List Index, together with the Species Habitat Index and the Biodiversity Intactness Index, but in their present form there is little coverage of freshwater and marine biodiversity.

A major global effort to document and thus to protect biodiversity was the Global Assessment Report on Biodiversity and Ecosystem Services issued by IPBES [17]. The IPBES approach moves beyond analysis of species presence and absence, and instead takes a whole ecosystem approach to monitoring biodiversity using what are termed Essential Biodiversity Variables (EBVs). These are measured to indicate both the rate of change as well as the deviation from natural. These include aspects such as ecosystem structure, function, community composition, species populations, species traits and genetic composition. Examples given in IPBES [17] illustrate that a great deal of information on most of the above is already available and could be used for contribution to the SDGs. The report documents species and ecosystem declines since the early Anthropocene, ramping up into the present era. While the details cannot be represented here, what is relevant is the extent, volume and diversity of the data and information that is available and has been used by the IPBES, although disaggregation of the data to national level would be a challenge. This highlights the rather scant approach adopted by the SDGs.

The Convention on Biological Diversity (CBD) Strategic Plan for Biodiversity 2011–2020 (including the Aichi biodiversity targets), sought the world's support for monitoring biodiversity and in the Aichi targets provided a greater suite of indices than is present in the SDGs. CBD have published a series of Global Biodiversity Outlook reports that document the state of global biodiversity in considerable detail making use of Aichi indicators and additional information. As the current CBD strategic plan comes to an end in 2020, the CBD started preparation for the Post-2020 Biodiversity Framework to take it to 2050 [22]. Given that there is wide recognition that the Aichi targets have not succeeded [86], this new framework takes a "theory of change" approach in an attempt to ensure that decisions and targets are met. This CBD report cautioned that goals for conserving and sustainably using biodiversity and achieving sustainability cannot be met by current trajectories, and biodiversity goals for 2030 and beyond may only be achieved through transformative changes across economic, social, political and technological factors. This includes for the Aichi Biodiversity Targets and the 2030 Agenda for Sustainable Development, which will not be achieved on the basis of current trajectories [22]. During 2020 the CBD Post-2020 process is geared to make deliberate linkages to the SDG indicators, but uptake of this by the IAEG will be necessary to succeed.

Butchart et al. [86] provide a number of reasons why the Aichi targets have not performed well on the global stage, suggesting four key issues—ambiguity (e.g., use of the word "sustainability" without any attendant value or description to describe a target), quantifiability (most indicators lacked quantifiable elements), complexity, and redundancy—as being at the center of the inability to gain a clear perspective on the status of biodiversity. They recommended for the future a smaller number of more focused headline targets that are specific, quantified, simple, succinct, and unambiguous. During development of the post-2020 program, the CBD circulated documents detailing the linkages between the SDGs and the CBD [87]. The stated objective of these documents were to strengthen these linkages "by ensuring coherence with the 2030 Agenda, the Global Biodiversity Framework can strengthen and advance the implementation and achievement of the SDGs. Additionally, the Sustainable Development Goals and their targets and associated indicators can serve as a reference to formulate global commitments in the Global Biodiversity Framework and its monitoring framework". What was omitted in this alignment were the actual methods and data, which if aligned, would ensure commonality of data and at the same time substantially reduce country monitoring effort. The two programs have substantial contributions to make towards biodiversity protection. However, by each maintaining its independent structure, there is a potential to overburden countries to the detriment of both programs and ultimately of sustainability. Nevertheless, by documenting how the CBD indicators could contribute to the SDG targets, this adds valuable input and demonstrates how programs from outside the SDG Agenda could usefully be incorporated in a formal way thus easing reporting requirements and streamlining data collection. Considerable guidance has been forthcoming, with for example

Tickner et al. [75] providing an Emergency Recovery Plan for freshwater ecosystems and biodiversity including indicators that could support the SDGs and the CBS monitoring.

### 4.5. Beyond the SDGs for Natural Resources

The uptake of the SDG Agenda by most countries of the world is unprecedented [23] and should become its defining attribute, provided that the data that are produced are meaningful. The diversity of the SDGs, with data that cover the whole globe, enables trajectories of social, economic and environmental indicators to be interlinked, potentially providing a synthesized picture to society that is more likely to be understood and thus to engender a response. It is also a strength of the SDGs that they are not overwhelmed by complexity even though there are >240 indicators, but these are distributed widely across the field of sustainability so need not create confusion. However, the deficiencies of the SDG Agenda are beginning to show, with a key issue being the lack of interdependency between natural resource including biodiversity, ecosystem services and sustainable development [88].

While Agenda 2030 and the SDGs are globally recognized for driving the sustainability of the planet forward, there are many other initiatives that are more focused and produce a greater level of detail than does Agenda 2030. These include the IPBES (biodiversity), the Convention on Biological Diversity and the post-2020 monitoring initiative (biodiversity and ecosystems), the Living Planet Index (WWF and ZSL—trends in species abundance), World Water Quality Assessment (WWQA—water quality), GEMStat (UNEP—water quality), GEMSAir (UNEP—air quality), Regional Seas (UNEP—oceans), IPCC (air and climate change), World Ocean Assessment (UNEP—oceans), Transboundary Water Assessment Program (TWAP—rivers, groundwater), Ramsar Convention on Wetlands of International Importance (wetland extent and condition), the System of Environmental Economic Accounting (SEEA), and many others. The linkages of these many programs to the SDGs are at best speculative. An initiative by the UN WCMC, the System of Environmental Economic Accounting (SEEA), reviewed the overlaps between the SEEA, the SDGs and a number of other initiatives to see whether the SEEA could provide the information required by the SDG Agenda [89]. They also reviewed other ecosystem monitoring programs (IPBES, CBD, etc.) and documented their overlaps with the SEEA and the SDGs. The SEEA approaches monitoring through the portal of SEEA Accounts, the first of which is ecosystem condition. They measure the area of ecosystems by type and the biophysical characteristics that help understand the condition of the ecosystems, and follow that with a number of economic evaluations of ecosystem services. They document for example that for SDG 6.6.1 on water-related ecosystems, there are conceptually equivalent indicators in the SEEA (Ecosystem Extent/Land Cover Account and SEEA Water Account), Aichi (AT 5.5.3 and AT 5.5.1), Ramsar (R 8.6), BIP (BIP B.1) and IPBES (IPBES H.10) where data could be shared between these initiatives. An important limitation of the SEEA approach, however, is that it is through a lens of economics, not necessarily what is needed for documentation of natural resources.

It is proposed here that these multiple programs, which provide both indicators and some level of monitoring and reporting, provide an invaluable resource that is measuring sustainability through a particular lens and that this information and data should not go wasted in development of a global perspective of sustainability, acknowledging that it may not contribute to country reporting. By integrating elements of these data directly into the SDGs, this could facilitate a more holistic evaluation of sustainability, overcoming many of the weakness with regard to natural resources shown here.

Table 3 provides just an overview of the detailed assessment of some deficiencies in the natural resource related SDG indicators provided above, identifying the key missing indicators that should either be developed for inclusion in the SDGs, or deliberate efforts made to incorporate data from other global programs as discussed above.

**Table 3.** Summary of inadequacies of SDG indicators that support quantification and protection of natural resources and proposals for what is missing. Details on these are available in Tables 1 and 2.

| Natural Resource | Summary of Inadequacies in Resource Protection | Key Missing Indicators |
|---|---|---|
| Land | Most indicators focus only on select resources, e.g., agricultural soil, forest area, greening of mountains, etc., and do not give a comprehensive evaluation of all landscapes. The land degradation indicator is most comprehensive but does not include all necessary perturbations. There is no comprehensive measure of the state of land ecosystems. | Several aspects of land degradation, e.g., soil condition, nutrients and fertility, salinization and desertification; vegetation cover of flat lands and wetlands; consumption of non-urban land; extent of different ecosystem types; the health of terrestrial ecosystems. |
| Water | Freshwater and marine water quality by necessity has limited variables that reduces assessment certainty. Quantity of water is well covered, as is spatial extent of freshwater ecosystems, but there is a need to disaggregate natural from artificial water bodies and wetlands. There is no comprehensive measure of the state of freshwater and marine ecosystems. | Many water quality variables, water stress during the dry season, quantities of water in wetlands, snow and ice and soil, extent of different water-related ecosystem types, discontinuity of river ecosystems, environmental flows, and ecosystem health in both marine and freshwater. |
| Air | Important air pollutants are missing, as is a deliberate link to climate change. | Additional pollutants, e.g., nitrogen dioxide, and sulfur dioxide; air temperature with more explicit linkages to the IPCC. |
| Biodiversity | Limited biodiversity assessments are included that form part of other indicators, e.g., fish stocks, forests and mountains, etc. There is limited information via the Red List but there is no deliberate biodiversity assessment. | Abundance and distribution of non-threatened species, genetic material of non-agricultural species, freshwater fisheries, and a comprehensive biodiversity assessment of all ecosystems. |

## 5. Conclusions

Assessment of sustainability is the core of the SDGs, and this paper has addressed just the natural resource component of sustainability. All four natural resources are covered by SDG indicators even if sometimes concealed within complex indices. Two major weaknesses exist and these are that (1) the health or state of land, water and air ecosystems are not included other than in a few partial cases; and (2) biodiversity is not included in any comprehensive form and the indicators provided will not be able to provide a comprehensive evaluation. These two weaknesses represent a challenge for the assessment of sustainability as a whole, and reveal that the SDGs have not fully embraced the value of ecosystems and biodiversity in the evaluation of sustainability. Ecosystems are now well accepted to be at the core of the interaction between humanity and the environment (MEA, TEEB, and IPBES), and biodiversity is at the core of the functioning of ecosystems (IPBES). Thus, the weakness of these two could be of crucial significance to the SDG Agenda. While the IAEG has shown little appetite to expand the indicators beyond just a few additions, the Convention on Biological Diversity (CBD) has proactively taken steps to identify the linkages between the very detailed CBD indicators and the SDGs, in anticipation that adoption at some level by the SDGs would strengthen this assessment. In the absence of uptake by the SDGs, this would turn the focus back towards the CBD to more successfully report on the sustainability of biodiversity, to the detriment of Agenda 2030.

Besides the definitions of the SDG indicators, their implementation is necessary at a global and country level before they can become effective. Two issues emerge here—(1) countries and also global reports tend to prioritize social and development indicators and give less attention to resource indicators [18,19,90] and the environment as a whole [91], and (2) countries are not willing and/or able to collect the data on natural resources necessary to make a country or global assessment of sustainability a possibility. For example, the global SDG 6 report on water concluded that the world was "not on track" to meet the SDG targets, and this was all the more concerning as, for many of the water resource indicators, no report was possible due to a lack of data [92]. Similarly, progress was reviewed by UN Environment where it was reported that 68% of the environmentally orientated indicators did not have enough data to be reported on [32]. This signals the delicate balance between the complexity of the SDG Agenda and the appetite or willingness of countries to embrace the SDGs. Dickens et al. [93] outlined how countries should be setting deliberate targets to direct monitoring programs at a country level, following the requirement of Agenda 2030 to do so.

This paper identifies many weaknesses in the monitoring of natural resources particularly in relation to the health of ecosystems and also biodiversity, while acknowledging that much of these data exist in parallel monitoring programs that operate at a global level. These global programs, however, may also not be implementable at a country level, which is a necessary condition of the SDGs. Overt linkages between the SDGs and these other programs could be the most pragmatic way of ensuring that the evaluation of sustainability is more holistic, but in order to ensure that the data are included in SDG reporting, it is proposed that data from these programs are directly incorporated into SDG evaluations of sustainability.

**Author Contributions:** Conceptualization, C.D.; methodology, C.D. and B.N.; validation, M.M., D.T., P.P. and I.J.H.; formal analysis, C.D.; M.M., D.T., P.P. and I.J.H.; investigation, C.D.; data curation, B.N.; writing—original draft preparation, C.D. and B.N.; writing—review and editing, C.D., M.M., D.T., P.P. and I.J.H.; visualization, C.D.; supervision, C.D.; project administration, C.D.; funding acquisition, M.M. All authors have read and agreed to the published version of the manuscript.

**Funding:** This research was funded by the Water, Land and Ecosystems (WLE) program that forms part of the CGIAR (see acknowledgements), Project Code P457: *Support to the roll-out of SDG monitoring and reporting by countries enhancing food security and sustainable use of water.*

**Acknowledgments:** The International Water Management Institute (IWMI) and the CGIAR Research Program on Water, Land and Ecosystems (WLE). The CGIAR Research Program on Water, Land and Ecosystems (WLE) combines the resources of 11 CGIAR centers, the Food and Agriculture Organization of the United Nations (FAO), the RUAF Foundation, and numerous national, regional and international partners to provide an integrated approach to natural resource management research. WLE promotes a new approach to sustainable intensification in which a healthy functioning ecosystem is seen as a prerequisite to agricultural development, resilience of food systems and human well-being. This program is led by the International Water Management Institute (IWMI) and is supported by CGIAR, a global research partnership for a food-secure future.

**Conflicts of Interest:** The authors declare no conflict of interest. The funders had no role in the design of the study; in the collection, analyses, or interpretation of data; in the writing of the manuscript, or in the decision to publish the results.

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
