# Peer review of "Evaluating the Global State of Ecosystems and Natural Resources: Within and Beyond the SDGs"

_sustainability, doi:10.3390/su12187381_

Round 1

Reviewer 1 Report

The question addressed in the research is interesting.

The text is clear and easy to read.

The conclusions are consistent with the arguments presented.

I have just a few comments

2.Materials and Methods

Lines 154 – 158 - Explain and justify the choice of criteria [1], [2] and [3]

Line 155 – units must be in SI (Mm3.annum-1)

Author Response

Many thanks for the review. 

Lines 154 – 158 - Explain and justify the choice of criteria [1], [2] and [3]

  • The following change was made: Data were gathered by seeking out those indicators that provide one of two perspectives related to resource protection; firstly a quantifiable measure of some aspect of the natural resources, that ensures that the resource is evaluated quantitatively thus enabling awareness of how much remains; and secondly those indicators that put in place measures that would directly support protection of the quantity of resources.

Line 155 – units must be in SI (Mm3.annum-1)

  • Changed to Mm3/a or mg/L

Reviewer 2 Report

Although it is not yet perfect in its presentation, it is highly relevant. We normally state that the paper is very actual as a matter of politeness, but this time I really mean it. SDGs are a most ambitious global plan to make the Earth Sustainable. Although very limited in their efficiency, we must understand that “wild problems” (as sustainability) cannot be cured as a house is built – with exact planning and little surprises during implementation. SDGs are much more specific than MDGs, not to mention the UN Declaration if Universal Human Rights shortly after WW2. But we must see this as a natural development procedure. I think that articles like this will contribute to the understanding of benefits and limitations of global goal settings and an “up-tuning” of the next generation of global goals after 2030. I just want to emphasize that the authors do not fall to the attempting and obvious mistake of depreciating and underpositioning the SDGs – although they would have some background (as at the first sight they have more than sufficient background for that; SDGs seem to be very poor on measurable natural resource development). In Hungarian we have a saying, but the authors do not  fall into this trap to “throw the baby out with bathwater”.

However, there are some room for development:

  1. MINOR: There are some minor spelling and formatting mistakes, e.g. line 133 speaks about the “heath of the planet”. Literature reference numbers, when used for the second time (especially 23, the UN SDG basic document) are not hyperlinked correctly.
  2. MAJOR: More importantly, “I sense a disturbance in the Force” ;-) when presenting the triple clustering in lines 155-157. It is not smart to use the same format [1] … [3] as with literature reference, try i.) … iii) or something else! Also I am confused with the tier approach of IAEG, which is heavily relied and referred to hereafter. And we face a second hardly understandable signaling: 1-11-111. I would avoid binary numbering, unless really necessary. In the age of infographics and quick reading I strongly recommend to put more energy into presenting results in a simple and visual form in tables 1-3. And please explain clearly the commonalities and differences (if any) of your 3 clusters and that of IAEG!!! Especially table one is poorly presented, it reaches over 4 pages without a repeated header, etc. Figure one needs some rethinking as well: missing last sentence in the legend, number of indicators not specified only percentages, etc.
  3. MAJOR: If I were you, my main recommendation to UN to develop efficiency and scientific soundness (adding to yours, not omitting or substituting them) would be to use the pressure-state-response approach from the environmental performance evaluation of companies, as specified by the ISO 14031 standard. Back almost two decades ago, one of the major innovations of environmental performance evaluation through indicators, or the ISO 14031 standard respectively was that it does not only focus on operational performance indicators (OPIs), but raises other two categories: management performance indicators (MPIs) and environmental condition indicators (ECIs). It is a main logical strong point at the same time: the load on environment (OPIs) generates changes in its state (ECIs), to decrease these we take measures (MPIs). This is the pressure – state – response From your results it seems that UN SDGs are dominated by MPIs, without a proper link to ECIs and OPIs. (More detail e.g. in https://www.researchgate.net/publication/247850114_Evaluation_of_Environmental_Performance_of_Companies)
  4. MINOR: The virus pandemic should be mentioned, at least a second time, not only in the emission situation. Does this change the overall picture? More understanding of need for sustainability, coordinating role or disappointment in WHO/UN, changing priorities?

Author Response

Many thanks for the detailed review.  Here are some comments in response.

Minor spell check required

  • Spell check carried out

Although it is not yet perfect in its presentation, it is highly relevant. We normally state that the paper is very actual as a matter of politeness, but this time I really mean it. SDGs are a most ambitious global plan to make the Earth Sustainable. Although very limited in their efficiency, we must understand that “wild problems” (as sustainability) cannot be cured as a house is built – with exact planning and little surprises during implementation. SDGs are much more specific than MDGs, not to mention the UN Declaration if Universal Human Rights shortly after WW2. But we must see this as a natural development procedure. I think that articles like this will contribute to the understanding of benefits and limitations of global goal settings and an “up-tuning” of the next generation of global goals after 2030. I just want to emphasize that the authors do not fall to the attempting and obvious mistake of depreciating and underpositioning the SDGs – although they would have some background (as at the first sight they have more than sufficient background for that; SDGs seem to be very poor on measurable natural resource development). In Hungarian we have a saying, but the authors do not  fall into this trap to “throw the baby out with bathwater”.

However, there are some room for development:

MINOR: There are some minor spelling and formatting mistakes, e.g. line 133 speaks about the “heath of the planet”. Literature reference numbers, when used for the second time (especially 23, the UN SDG basic document) are not hyperlinked correctly.

  • Line 133 - Corrected the spelling of “health” and changed to “health status”
  • Comprehensive spelling and grammar check done.
  • Reference links corrected to reference 23 (lines 111, 104, )

MAJOR: More importantly, “I sense a disturbance in the Force” ;-) when presenting the triple clustering in lines 155-157. It is not smart to use the same format [1] … [3] as with literature reference, try i.) … iii) or something else!

  • Line 155-157 Done as recommended – changed to i) etc

Also I am confused with the tier approach of IAEG, which is heavily relied and referred to hereafter.

  • We did not want to detail the IAEG tiers as these are only of partial interest. Thus I have removed the IAEG tiers from the Tables 1&2 and reduced the text (see below)

And we face a second hardly understandable signaling: 1-11-111. I would avoid binary numbering, unless really necessary.

  • Changed to 1,2,3

In the age of infographics and quick reading I strongly recommend to put more energy into presenting results in a simple and visual form in tables 1-3. And please explain clearly the commonalities and differences (if any) of your 3 clusters and that of IAEG!!! Especially table one is poorly presented, it reaches over 4 pages without a repeated header, etc. Figure one needs some rethinking as well: missing last sentence in the legend, number of indicators not specified only percentages, etc.

  • Figure 1 has been redrawn, numbers of indicators included, colours included
  • Tables 1&2 both refined – reduced text size to make them smaller and changed margins etc. Removed all reference to Tiers as in the end they were all available for implementation thus a reference to this is added in the text below the Table 1.   The following text is added:
  • Line 183 - 186 - The SDG indicator methods are in a continual state of development, categorised by the IAEG into Tier 3 (under development), Tier 2 (ready but not operational) and Tier 1 (fully operational in many countries). According to the IAEG in 2020 [36] the Tier status of all of the indicators in Tables 1&2 were either Tier 1 or 2 and thus all were available for implementation.

MAJOR: If I were you, my main recommendation to UN to develop efficiency and scientific soundness (adding to yours, not omitting or substituting them) would be to use the pressure-state-response approach from the environmental performance evaluation of companies, as specified by the ISO 14031 standard. Back almost two decades ago, one of the major innovations of environmental performance evaluation through indicators, or the ISO 14031 standard respectively was that it does not only focus on operational performance indicators (OPIs), but raises other two categories: management performance indicators (MPIs) and environmental condition indicators (ECIs). It is a main logical strong point at the same time: the load on environment (OPIs) generates changes in its state (ECIs), to decrease these we take measures (MPIs). This is the pressure – state – response From your results it seems that UN SDGs are dominated by MPIs, without a proper link to ECIs and OPIs. (More detail e.g. in https://www.researchgate.net/publication/247850114_Evaluation_of_Environmental_Performance_of_Companies)

  • Very useful perspective to add – the following changes have been made….
  • Lines 187-189 - Tables 1&2 have been structured according to guidelines of ISO 14031, which divides indicators into three categories; environmental condition indicators (ECIs) which are in Table 1; operational performance indicators (OPIs) and management performance indicator (MPI) are in Table 2.
  • Lines 190-194 - The information in Table 1 reflects only those indicators that directly and deliberately set out to quantify some aspect of the natural resource, and are thus a measure of environmental condition (ECI) and may be affected by over-utilization. In some indicators this may be included in an index where the natural resource measure may be obscured (e.g. 15.4.2 – the Mountain Green Cover Index), the indicator thus requiring disaggregation. 
  • Lines 195-205 - The indicators presented in table 2 do not themselves contain any quantification of natural resources and thus are not able to reflect their actual status. However, these indicators may provide a basis for protection of natural resources by providing enabling conditions, for example indicators 14.6.1 does not monitor any fish species but rather country implementation of the Code of Conduct for Responsible Fisheries and related instruments (CCRF) which encourages the conservation and protection of fish stocks through combating illegal fishing and over exploitation. In the same manner indicator 6.5.1 does not monitor any water quality or quantity parameters but by monitoring country commitments to implementation of policies, laws, institutional arrangements, budgeting and financing and strategies for water resources development, protection of water quality may be achieved. Table 2 is thus a mix of operational performance indicators (OPIs) and management performance indicator (MPI).

MINOR: The virus pandemic should be mentioned, at least a second time, not only in the emission situation. Does this change the overall picture? More understanding of need for sustainability, coordinating role or disappointment in WHO/UN, changing priorities?

  • While the virus pandemic is important, its links to natural resources are only just now emerging. We feel that this would be just one of many examples of how natural resource abuse is impacting on civilisation, climate change being arguably even more significant.  We have chosen to avoid going into discussion about the so-what of exploitation, as this would require considerable lengthening of the paper. 

Reviewer 3 Report

(1) There have been some other similar articles. What is the difference with those published papers?

(2) Some figures can be improved. For example, different colors can be used for the pie chart.

(3) I didn't see any methodology part. Should this article be a review paper?

(4) It's expected to see more discussions about future chanllenges.

Author Response

Many thanks for the review.  Here is our response.

) There have been some other similar articles. What is the difference with those published papers?

  • Reference to many similar articles has been made (e.g. Wagernackel, Reference 37, 38, 39, 32) however none of these takes the in-depth approach taken here where the indicators were evaluated for their potential performance with regard to natural resources. This is indicated in the early part of the discussion.  We feel that this is sufficient.

(2) Some figures can be improved. For example, different colors can be used for the pie chart.

  • Figure 1 has been redrawn

(3) I didn't see any methodology part. Should this article be a review paper?

  • The Method section begins at row 147. It is short but sufficient.

(4) It's expected to see more discussions about future challenges.

  • There was concern that the paper at 13,000 words was already excessively long. We feel that this conclusions provide sufficient suggestion about the way forward:
  • We point out and document the weaknesses of the SDG indicators
  • The implementation of SDG indicators is prioritised by countries to development indicators and they also do not collect sufficient data on natural resources
  • A key conclusion of the paper was that while the SDG indicators may not be sufficient – there are a number of global monitoring programmes already in place (IPBES, CBD etc) that are already providing a lot of information. The SDGs could be structured to absorb conclusions arising from these programmes and thus to reflect sustainability more broadly. 
  • We feel that given the constraints on length – and the detail already offered, that we need not add to the discussion.

Round 2

Reviewer 3 Report

The author has improved the manuscript.